# Defect Detection Methods for Industrial Products Using Deep Learning Techniques: A Review

**Alireza Saberironaghi [1], Jing Ren [1] and Moustafa El-Gindy [2,*]**

1   Department of Electrical, Computer and Software Engineering, Ontario Tech University, Oshawa, ON L1G 0C5, Canada

2   Department of Automotive and Mechatronics Engineering, Ontario Tech University, Oshawa, ON L1G 0C5, Canada

*   Correspondence: moustafa.el-gindy@ontariotechu.ca; Tel.: +1-905-721-8668 (ext. 5718)

**Abstract:** Over the last few decades, detecting surface defects has attracted significant attention as a challenging task. There are specific classes of problems that can be solved using traditional image processing techniques. However, these techniques struggle with complex textures in backgrounds, noise, and differences in lighting conditions. As a solution to this problem, deep learning has recently emerged, motivated by two main factors: accessibility to computing power and the rapid digitization of society, which enables the creation of large databases of labeled samples. This review paper aims to briefly summarize and analyze the current state of research on detecting defects using machine learning methods. First, deep learning-based detection of surface defects on industrial products is discussed from three perspectives: supervised, semi-supervised, and unsupervised. Secondly, the current research status of deep learning defect detection methods for X-ray images is discussed. Finally, we summarize the most common challenges and their potential solutions in surface defect detection, such as unbalanced sample identification, limited sample size, and real-time processing.

**Keywords:** defect detection; surface defect detection; defect detection for X-ray images; defect recognition; deep learning

## 1. Terminology

- Support Vector Machine (SVM): an algorithm used in supervised learning for classifying and performing regression tasks.
- Region of Interest (ROI): an area within an image or video that is deemed particularly significant or relevant.
- Local Binary Patterns (LBP): a technique used in computer vision for extracting features and analyzing images.
- Reduced Coordinate Cluster Representation (RCCR): a method for representing and processing image data for object recognition that is efficient.
- Convolutional Neural Network (CNN): a neural network architecture commonly used for image and video processing tasks.
- Zero Defect Manufacturing (ZDM): a strategy to eliminate defects in the manufacturing process and improve quality.
- Deep Neural Network (DNN): a neural network architecture with multiple layers, commonly used for image recognition and natural language processing tasks.
- MobileNet Single Shot MultiBox Detector (MobileNet-SSD): a lightweight convolutional neural network that is designed for real-time object detection on mobile and embedded devices.
- Fully Convolutional Network (FCN): a neural network architecture used for semantic segmentation tasks.
- Region-based Convolutional Neural Network (RCNN): a neural network architecture used for object detection tasks.

- Autoencoders (AEs): a neural network architecture used for unsupervised learning tasks such as dimensionality reduction and anomaly detection.
- Generative Adversarial Networks (GANs): a neural network architecture used for generative tasks such as image synthesis and image-to-image translation.
- Self-Organizing Map based (SOM-based): an unsupervised learning algorithm that organizes data into a 2D grid of clusters.
- General-purpose Annotation of Photos and Replica (GAPR) datasets: created by the German Pattern Recognition Association, is a collection of images specifically designed for the detection of texture defects.
- German Association for Pattern Recognition (DAGM) datasets: a collection of images specifically designed for the detection of textured surfaces.
- Northeastern University (NEU) datasets: created by Northeastern University, a collection of images of surface defects that includes six different types of defects.
- Convolutional Denoising AutoEncoder (CDAE): a type of autoencoder designed to remove noise from images.
- Non-Destructive Testing (NDT): a method of evaluating the properties of a material, component, or system without causing damage.
- VGG: VGG is a pioneering object-recognition model that can have up to 19 layers. Created as a deep CNN, it surpasses other models on many tasks and datasets apart from ImageNet. VGG is still a widely used architecture for image recognition today.
- Mean Average Precision (mAP): a metric used to evaluate the performance of object detection models, that calculates the average precision across different classes and object instances.

## 2. Introduction

Several factors affect the quality of manufactured products during the manufacturing process, including poor working conditions, inadequate technology, and various other factors. Among product defects, poor product quality is most visible in surface defects. Therefore, detecting product surface defects [1] ensures a high qualification ratio and reliable quality.

A defect is generally defined as an absence or area that differs from a normal sample. Figure 1 compares normal samples with defective samples of industrial products.

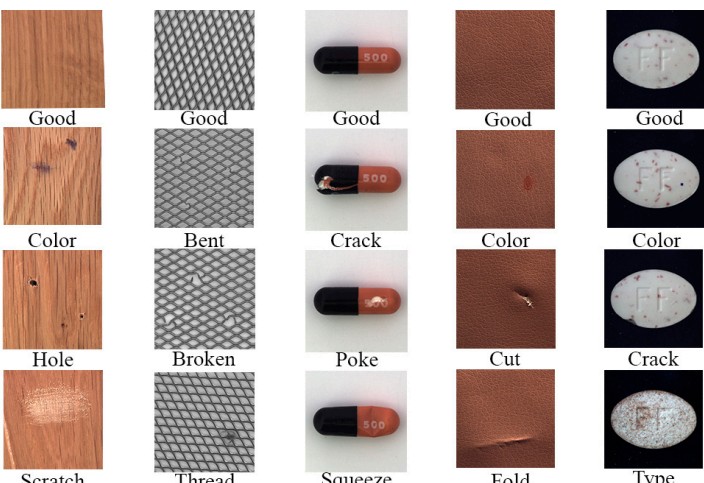

**Figure 1.** Normal samples of industrial products are compared to defective samples. The first row contains good samples, and the second, third, and fourth rows contain defective samples. The first, second, third, fourth, and fifth columns display wood, grid, capsule, leather, and bill, respectively, and there are three types of defects listed below the image.

In the past, identifying defects was carried out by experts, but this process was not efficient. One major reason for this was that human subjectivity greatly affected the detection results. Additionally, human inspection alone cannot meet the need for real-time detection, and thus, it is not able to fulfill all the necessary requirements.

A significant amount of time has been dedicated to using traditional methods to detect surface defects. When differentiation exists between the defect color and the background, traditional image processing methods can perform well. Traditional methods in terms of the product's features can be categorized into three types: texture-based features, color-based features, and shape-based features.

Several studies have used specialized techniques for detecting surface defects. In color-based feature, for instance, literature [2] proposed a technique involving the use of a percentage of the feature of color histogram and a vector texture feature to classify image blocks to detect surface defects on wood; this method has been proven effective by experiments, especially with defects involving junctions. In Figure 2, the method results are shown.

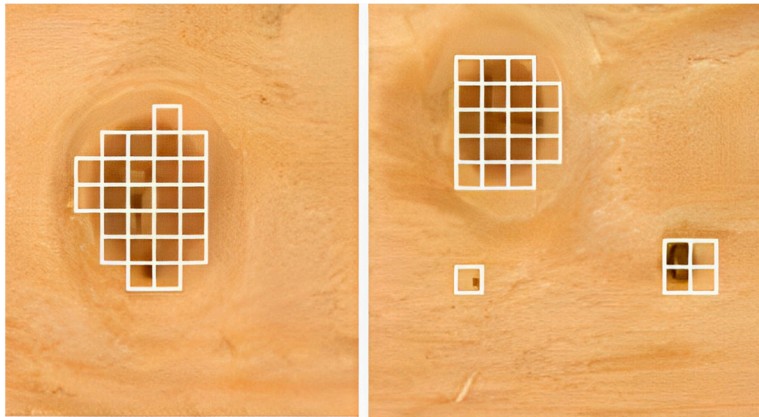

**Figure 2.** An example of the result of wood defect detection using the presented technique in [2].

Research conducted in literature [3] employed cosine similarity to verify the validity of the periodic law in magneto-optical images by utilizing the color moment feature. This method was successful in identifying the appropriate magneto-optical image for detecting and locating welding defects. Literature [4] describes a two-step technological process for SVM-based and color histogram-based defect detection in particle boards, followed by localization of defects using smoothing and thresholding. According to literature [5], color moment features and FSIFT features were merged based on their magnitude of influence for the purpose of resolving the tile surface defect problem not being adequately described by a single feature.

In terms of shape-based feature methods, literature [6] proposed a method of detecting cutting defects on magnetic surfaces. In this method, the image of the magnetic surface is reconstructed using the Fourier transform and Hough transform, and, in order to obtain defect information, the gray difference between the original image and the reconstructed image is compared. A method for identifying defects on bottle surfaces was presented in reference [7]. This method includes a step for extracting regions of interest, where the boundary line of the light source is determined using a fast Hough transform algorithm. In [8], global Fourier image reconstruction and template matching were proposed as a method for detecting and locating small defects in aperiodic images. Literature [9] described how to detect surface defects on small camera lenses using Hough transforms, polar coordinate transforms, weighted Sobel filters, and SVM algorithms. Different types of defects were detected in several test images. In Figure 3, red highlights are used to indicate defects such as stains, scratches, and dots.

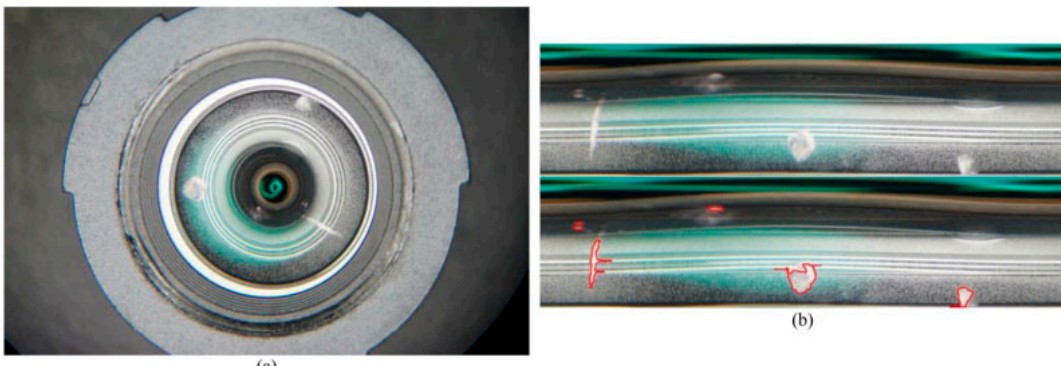

**Figure 3.** A camera lens with several defects: (**a**) original image and (**b**) converted result based on inspection result and polar coordinate transformation [9].

In the texture-based feature method, for example in [10], a multi-block local binary pattern (LBP) algorithm has been improved. In addition to having the simplicity and efficiency of LBP, this algorithm ensures high recognition accuracy by varying the block size to describe defect features. According to the experiment, the method has the speed to meet online real-time detection requirements (63 milliseconds/image), outperform the widely used scale-invariant feature transform (SIFT), speed up robust features (SURF), and gray-level co-occurrence matrix (GLCM) algorithms for recognition accuracy (94.30%), demonstrating that MB-LBP can be used to detect images in real time online. Literature [11] used a fuzzy model that was based on extracting GLCM features and processed it using MATLAB. The model took in three variables as inputs: autocorrelation, square root of variance, and the number of objects. Using fuzzy logic on ceramic defects, the accuracy of the ceramic inspection process with a light intensity of 300 lx, camera distance of 50 cm, and a 1.3 MP or 640 × 480 pixel image size was determined using the training data of 96.87%, and the accuracy of the real-time system was 92.31%. According to literature [12], features such as Reduced Coordinated Cluster Representation (RCCR) are used to form a one-class classifier. An algorithm based on texture periodicity estimates the primitive unit size of defect-free fabrics during the training phase. After splitting the fabrics into samples of one unit, RCCR features are used in a one-class classifier to learn their local structure. In [13], morphological filters are used to detect defects on billet surfaces in order to distinguish them from scales. With the help of morphological erosion and dilation techniques with repetition, the image is converted into a binary image by using morphological top-hat filtering and thresholding. The detection efficiency of the proposed algorithm is evaluated using real billet images to evaluate its performance. The proposed algorithm is found to be effective and suitable for analyzing billet images with scales in experiments. According to literature [14], the GLCM is defined as the fabric image's characteristic matrix. To distinguish defect-free from defective images, Euclidean distance is used and, in order to determine the pattern period, the autocorrelation function is used. In this paper, the authors discussed two GLCM parameters in relation to Euclidean distances. Furthermore, in addition to being concise and objective, Euclidean distances have the advantage of being reliable and objective for defect detection. According to the algorithm's tests, it is not only accurate, but also more adaptable to yarn-dyed fabrics with short organization cycles. Table 1 summarizes recent applications of machine learning algorithms for surface defect detection in industrial products, categorized by texture, color, and shape features. Table 2 compares the strengths and weaknesses of feature-based methods for detecting surface defects, including accuracy, computational efficiency, and robustness. These tables provide an overview of the diversity of approaches and key factors affecting performance in the field of surface defect detection.

**Table 1.** Recent applications using machine-learning-based vision algorithms for detecting surface defects in industrial products, categorized into three categories based on texture, color, and shape features.

| Approach | Reference | Feature | Target | Performance |
|---|---|---|---|---|
| Texture-based | [11] | Gray level co-occurrence matrix | Ceramic | Recognition rate: 92.31% |
| | [13] | Mathematical morphological | Billet | Accuracy: 87.5% |
| | [15] | Fractal model | Steel | Accuracy: 88.33% |
| | [16] | Gabor filter | Steel billet | Thin crack: 91.9% and corner crack: 93.5% |
| Color-based | [3] | Bivariate color histogram | Particleboards | Can effectively detect and localize defect |
| | [17] | Color coherence vectors combined with texture features as a basis | NWPU-RESISC45 data sets | Accuracy: 96.66% |
| | [18] | Color histogram | Cementitious materials | ERT can be efficient for situ monitoring and defect detection of cement mortar |
| Shape-based | [6] | Fourier image | Magnet | Can automatically detect surface-cutting defects in magnets |
| | [8] | Comparison of the whole Fourier spectra between the template and the inspection image | Non-periodical pattern images | Can detect various types of non-repeating patterns in the electronic industry, even those as small as one pixel wide, making it useful for identifying defects |
| | [9] | A circle Hough transformation, weighted Sobel filter, and polar transformation | Compact camera lens | Able to identify defects in complicated circular inspection areas and has been proven to be highly effective |

**Table 2.** An overview of the strengths and weaknesses of various feature-based methods for detecting surface defects in industrial products.

| Approach | Reference | Method | Strengths | Weaknesses |
|---|---|---|---|---|
| Texture-based | [10] | Multi-block local binary pattern (LBP) algorithm | High recognition accuracy and meets online real-time detection requirements; robust to rotation and scaling; fast processing time | Does not perform well with defects that do not involve texture changes; may not be able to detect defects with low contrast; |
| | [11] | Fuzzy model based on GLCM extraction | Can be useful for detecting defects in images with low contrast or noise, where other methods may fail | Not as good at detecting defects that have a very different texture than the one used to train the model; may not be as accurate as deep learning-based methods, which can learn from data and adapt to new types of defects |
| | [12] | Reduced Coordinated Cluster Representation (RCCR) | Good at detecting defects with high precision, as it is able to extract features of the defects and identify them; good at detecting defects in images with low contrast or noise, as it is able to extract features that are robust to these challenges | It is limited to detecting specific types of defects (based on the specific clustering method and feature extraction technique used), which can make it less suitable for more complex or varied defects |

**Table 2.** *Cont.*

| Approach | Reference | Method | Strengths | Weaknesses |
|---|---|---|---|---|
| Color-based | [2] | Color histogram and vector texture feature | Proven to be effective with defects involving junctions; able to handle multiple input features | Not be suitable for detecting defects in textures with complex patterns; may not work well for defects that do not involve changes in color |
| | [3] | Cosine similarity and color moment feature | A robust method for comparing similarity between images, which can be useful for detecting small defects that are difficult to see with the naked eye; are able to identify different types of defects with high precision, as they are able to extract features of the defects and identify them | May require additional preprocessing steps, such as image enhancement techniques, to improve their performance; do not have the ability to learn from data as compared to deep learning based methods, which can make them less adaptable to new types of defects or variations in the data |
| | [4] | SVM-based and color histogram-based | High accuracy rate; able to extract useful information from the color of an image, which can be useful for detecting defects that are based on color variations, such as stains or discolorations | May not perform well with other types of materials; may not be able to detect defects with low contrast |
| | [5] | Color moment features and FSIFT features | Successful in resolving tile surface defect problem not being adequately described by a single feature | May not perform well with defects that do not involve color changes; not be able to detect defects with low contrast |
| Shape-based | [6] | Fourier transform and Hough transform | Good at detecting periodic patterns, which can be useful for detecting defects in materials with repeating patterns, such as in fabrics or metals | Do not have the ability to learn from data as compared to deep learning based methods, which can make them less adaptable to new types of defects or variations in the data |
| | [7] | Fast Hough transform | Good at detecting linear features, such as cracks or scratches, in an image; good at detecting defects with high precision, as it is able to extract features of the defects and identify them. | Is not as good at detecting defects in images with low contrast or noise, which can make it less effective in some industrial applications; does not have the ability to learn from data as compared to deep learning based methods, which can make it less adaptable to new types of defects or variations in the data |
| | [8] | Global Fourier image reconstruction and template matching | Good at detecting small defects, such as scratches or cracks, in an image by reconstructing the original image from the Fourier domain | Limited to detecting specific types of defects (based on the specific templates or reconstruction of the Fourier domain), which can make them less suitable for more complex or varied defects |

Using only one feature or one class of features on industrial products is rarely sufficient because their surfaces typically contain a variety of information. Consequently, many features are used in combination in practical applications, making it difficult to detect defects. Additionally, feature-based approaches are highly effective when they detect defects in images with little or no variation, and when defects appear on surfaces in a consistent pattern. Considering the wide range of uncertainties in industrial settings, it is important to develop methods that are adaptable to such wide ranges of variations in defect intensity, shape, and size.

Deep learning models based on convolutional neural networks (CNN) have had a lot of success in various computer vision fields, such as recognizing faces, identifying pedestrians, detecting text in images, and tracking targets. Additionally, these models are used in a wide range of industrial settings for defect detection. This includes both

commercial and industrial applications, such as in the automotive industry for detecting defects in cars. The deep-learning-based surface defect detection software is employed in these settings to improve the efficiency and accuracy of the defect detection process.

Recently, several papers covering the latest techniques, applications, and other aspects have been published on deep learning in defect detection [19]. Literature [12] describes the different types of defects and compares mainstream and deep learning methods for defect detection. Various defect detection techniques are discussed in literature [20], including ultrasonic inspection, machine vision, and deep learning. Literature [21] focuses on the use of AI-enhanced metrology, computer vision, and quality assessment in the Zero Defect Manufacturing (ZDM) process. The study also highlights the use of IoT/IIoT technology as a means of supporting these tools and implementing AI algorithms for data processing and sharing. Literature [22] discusses deep learning methods for detecting surface defects, then discusses three critical issues related to small samples and real-time defects detection. In [23,24], the authors analyze and compare the benefits and drawbacks of the above methods. There are also defect detection surveys in several application domains, including fabric defects [25], corrosion detection [26], pavement defects [27], metal defect detection [28], and industrial applications [29]. The investigation shows that, in the field of surface defect detection of industrial products, there is currently a limited literature review on machine learning methods, and while some papers summarize the challenges and problems, the mentioned solutions are not systematic. The first section of this paper addresses the above issues by summarizing the research status on the detection of surface defects on industrial products using deep learning algorithms and then discusses the issues in the process of industrial surface defect detection, such as unbalanced sample identification problems, small sample problems, and real-time problems.

This paper is organized as follows. Section 3 provides an overview of deep learning methods for surface defect detection in industrial products from three perspectives, along with a common dataset for surface defect detection. In Section 4, we summarize the recent research status of deep learning methods for X-ray image defect detection. A discussion of the main problems and their solutions is provided in Section 5. In Section 6, a brief description of future research directions is provided and Section 7 concludes the paper with a conclusion.

## 3. Deep Learning Surface Defect Detection Methods for Industrial Products

Deep learning has become increasingly popular in the field of defect detection due to its rapid development. This section summarizes the state of research on inspection of industrial products for detecting surface defects. Learning-based approaches are classified as supervised, semi-supervised, and unsupervised. The performance of learning-based methods is best optimized when large datasets are provided. In particular, supervised techniques perform well when there are sufficient examples of each class in the dataset.

### 3.1. Supervised

Supervised detection requires large datasets of defect-free and defective samples labeled in a training set. Since all the training data is labeled, detection rates can be very high. It must be noted, however, that supervised detection may not always the most effective approach due to the imbalance of classes in the datasets. There are a number of datasets that supervised learning methods use, including the fabric dataset [30], rail defect dataset [31], and railroad dataset [32].

Deep neural networks and feature extraction and classification methods used in supervised methods differ in their structures. For example, detecting cross-category defects without retraining was proposed using a two-layer neural network in the literature [33]. Based on structural similarities between image pairs, the method learns differential features, which may result in some structural similarities among different classification objects. This method has been shown to be able to detect defects in different types of factories based on experiments in real factory datasets. Literature [34] suggests that the composition of kernels

is more important than the number of layers when it comes to detection results. To detect small defects and textures in surface images, it is necessary to use a sample image that is large enough for computational accuracy and reducing the cost of the network. ShuffleNet uses convolution of pointwise groups and channel shuffle as two new techniques to achieve this goal. Literature [35] proposes a novel in-line inspection system for plastic containers based on ShuffleNet V2. The system can be used to inspect images on complex backgrounds as well. In [36], they proposed ShuffleDefectNet, a deep-learning-based defect detection system that achieved 99.75% accuracy on the NEU dataset.

Reference [37] suggested that shallow CNN networks can be used to identify anomalies. To train the model, only negative images are used and the research employs full-size images. The argument is that it is not necessary to have full-size examples of both defective and defect-free samples, as the negative samples already have pixels that correspond to the defect-free regions. Based on the Fast R-CNN model, Faster R-CNN introduces a region proposal network (RPN), which enables an end-to-end learning algorithm. This leads to a near-costless regional recommendation algorithm that significantly improves the speed of target detection. Faster R-CNN was used in [38] to detect PCB surface defects, a new network was proposed combining ResNet50, GRAPN residual units, and ShuffleNetV2. Using a cascaded RCNN structure, as described in literature [39], the defect detection problem of power line insulators can be changed into a two-level target detection problem; the results are shown in Figure 4.

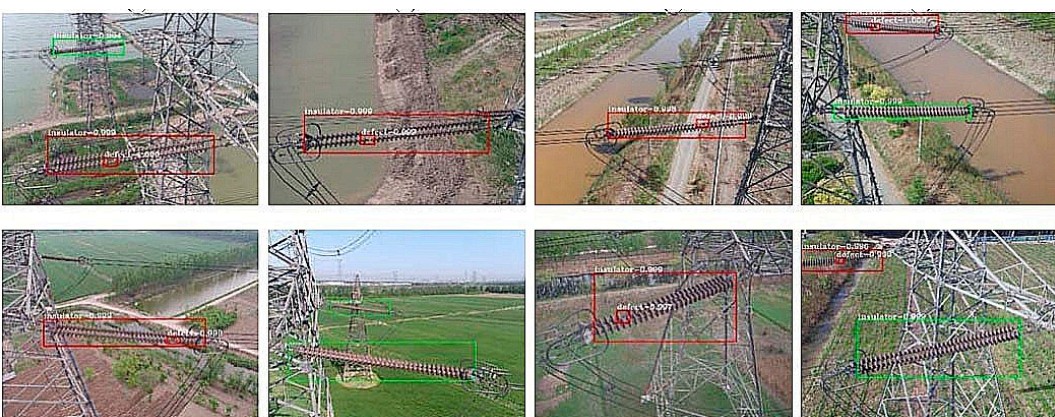

**Figure 4.** The results of insulator defect detection. The green box represents the non-defective insulator, and the red box represents the defective insulator [39].

In limited hardware configurations, MobileNet-SSD [9] improves real-time object detection performance. There is no need to sacrifice accuracy for the reduction of parameters in this network. An SSD network classifies regression and boundary box regression using various convolution layers. Translation invariance and variability are resolved in this model, resulting in good detection precision and speed. Object detection is effective when defects have regular or predictable shapes [40]. Additional preprocessing steps can be applied to more complex defect types. Fully Convolutional Networks (FCNs) use all convolutional layers as network layers; label maps can be directly derived using pixel-level prediction. To achieve accurate results, a deconvolution layer with larger data sizes is used. In literature [41], FCN and Faster R-CNN were combined to develop a deep learning model that could detect stains, leaks, and pipeline blockages in tunnels. A method for segmenting defects in solar cell electroluminescence pictures was presented in [42]. A defect segmentation map was obtained in one step by combining FCN with a specific U-net architecture.

*3.2. Unsupervised*

Research has begun to explore unsupervised methods to overcome the disadvantages of supervised methods. By learning the inherent characteristics of the input training data, the machine can learn some of its own characteristics and connections when there is no label information and automatically classifies the input training data based on the pattern of these unlabeled data [43]. It automatically classifies these unlabeled data based on inherent characteristics and connections between the data. Methods based on reconstruction and embedding similarity are the most commonly used to detect surface defects among unsupervised learning methods. Reconstruction-based methods such as autoencoders (AEs) and Generative Adversarial Networks (GANs) are most commonly used. Popular algorithms include PaDIM [44], SPADE [45] PatchCore [46], etc. In [47], an algorithm based on DBN was proposed for detecting defects in solar cells. Both training and reconstructed images were used as supervision data by the fine-tuning network of the BP algorithm. Literature [48] proposed a multi-scale convolutional denoising autoencoder with high accuracy and robustness that synthesizes the results of multiple pyramid levels.

A SOM-based detection method was proposed in [49] for determining the difference between normal and defective wood. The first stage involves detecting suspected defect areas, and the second stage involves separately inspecting each defect area. A detection method that uses GANs was proposed in reference [50]. The method is divided into two stages: first, a generative network and a learning mechanism based on statistical representation are used to detect new areas. In the second stage, defects and normal samples are directly distinguished using the Frechet distance. The solar panel dataset was used to test the method, and it achieved 93.75% accuracy.

A multiscale AE with fully convolutional neural networks has been proposed [51], in which each FCAE sub-network directly obtains the original feature image from the input image and performs feature clustering. Utilizing a fully convolutional neural network, the residual images were combined to create the defect image. PatchCore, introduced in literature [46], is a technique for identifying and isolating abnormal data in scenarios where only normal examples are available. It balances the need to retain normal context through memory banks of patch-level features extracted from pre-trained ImageNet networks and minimize computational time via coreset subsampling to create a leading system for cold-start image anomaly detection and localization that is efficient on industrial benchmarks. On MVTec, the algorithm demonstrated an AUROC of over 99%, while also being highly efficient in small training set scenarios. Literature [52] presented a GAN-based surface vision detection framework that uses OTSU to segment fusion feature response maps and fuses the responses of the three layers of the GAN discriminator. The framework has been proven effective on datasets of wood cracks and road cracks. As shown in Figure 5, ref. [53] proposed a GAN-based method for detecting strip steel surface defects, in which the generator G uses encoding and the hidden space features in the penultimate layer are fed into a SVM to detect defects. The test results on images provided by the Handan Iron and Steel Plant indicated good accuracy. It is more effective at detecting texture images; however, its accuracy still needs to be improved.

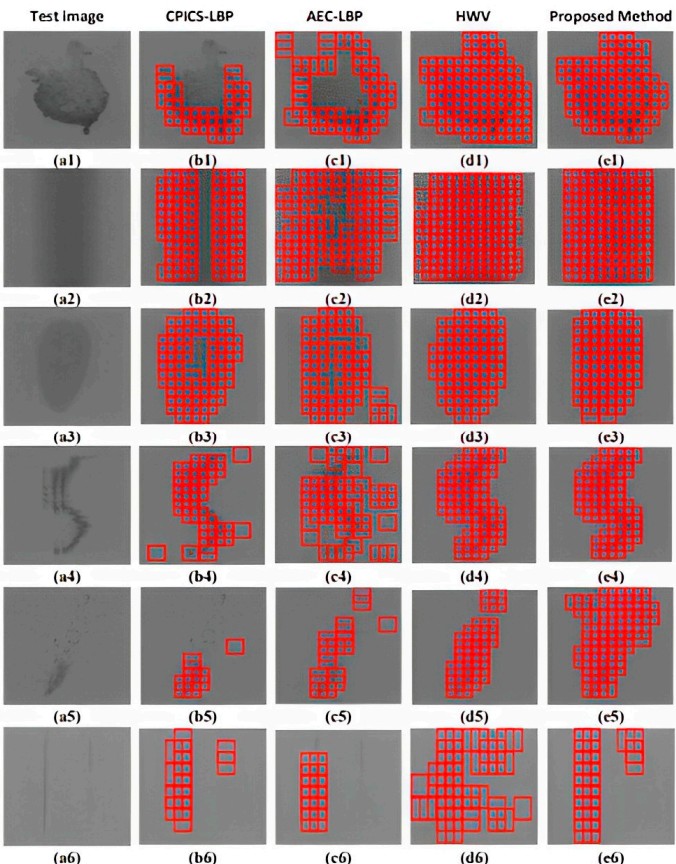

**Figure 5.** Presenting the results of experiments on six defect samples using four methods. The defect types are listed in the first column and include drops tar, shadow, floating, crush, pitted surface and scratch. The results from traditional manual feature extraction methods (CPICS-LBP, AEC-LBP, HWV and the proposed method in [53]) are shown in columns 2–5. The experiment compares the proposed method with current state-of-the-art methods in detecting strip steel surface defects.

### 3.3. Semi-Supervised

As a result of combining the properties of supervised and unsupervised methods, semi-supervised methods are developed. Only normal samples are used as training data for semi-supervised defect detection and a defect-free boundary is learned and set, and any samples outside the boundary are considered anomalous. Since there are few defective samples to be obtained, the method is extremely useful. Nevertheless, this method has lower accuracy in defect detection compared to supervised methods. Unlabeled sample data can be automatically generated by semi-supervised methods without manual intervention.

A framework for identifying defects in PCB solder joints was proposed in literature [54], which utilizes a combination of active learning and self-training through a sample query suggestion algorithm for classification. The framework has been demonstrated to improve classification accuracy while reducing the need for manual annotations. A semi-supervised model of convolutional autoencoder (CAE) and generative adversarial network is proposed in [55]. After training with unlabeled data, the stacked CAE's encoder network is retained and input into the SoftMax layer as a GAN discriminator. Using GAN, false images of steel surface defects were generated to train the discriminator. For the detection of steel surface defects, literature [56] developed a WSL framework combining localization networks (LNets) and decision networks (DNets), with LNets trained by image level labels and outputs a heat map of potential defects as input to DNets. Through the use of the RSAM algorithm to weight the regions identified by LNet, the proposed framework has been demonstrated to be effective on real industrial datasets. The application prospects for weakly supervised methods are also wide because the methods simultaneously com-

bine advantages of both supervised and unsupervised methods. There are few weakly supervised methods for detecting surface defects in industrial products. The literature [57] proposed a deep learning algorithm to learn defects from a variety of defect types with an unbalanced training sample pool for PCBA manufacturing products. In this method, an overall defect recognition accuracy of 98% is achieved in PBCA images using a novel batch sampling method and the sample weighted cost function.

A semi-supervised learning system that generates samples to detect surface defects was proposed according to the literature [58]. As part of the semi-supervised learning part, two CDCGAN and ResNet18 classifiers were used, and the NEU-CLS dataset was used to compare the two classifiers. In this way, supervised learning and transfer learning are both shown to be inferior to the method. A convolutional neural network structure based on residual network structures was proposed in [59] by stacking two layers of residual building modules together, resulting in a 43-layer convolutional neural network, while at the same time by appropriately increasing the network width; a more balanced network depth and network width can be obtained and accuracy can be improved. The network structure shows good performance on the DAGM, NEU steel, and copper clad plate datasets. Table 3 provides an overview of recent research in surface defect detection, including classifications of targets and Table 4 evaluates the strengths and weaknesses of deep learning techniques for detecting surface defects in industrial products, including accuracy, computational efficiency, and robustness. These tables give a comprehensive understanding of current research and the considerations for using deep learning in surface defect detection. Table 5 lists a selection of commonly used datasets for training and testing algorithms for detecting surface defects in industrial products. The datasets are classified based on the type of industrial products they are intended for. This information is useful for researchers and practitioners looking for suitable datasets for their work in the field of surface defect detection.

**Table 3.** An overview of recent research publications as well as classifications based on targets.

| Reference | Year | Method | Target | Performance |
|---|---|---|---|---|
| [46] | 2022 | PatchCore | MVTec benchmark datasets, the ShanghaiTech Campus dataset (STC), and the Magnetic Tile Defects dataset (MTD) | Demonstrated a high level of performance on the MVTec dataset with an AUROC of over 99% and a particularly strong ability to perform well with small training sets |
| [60] | 2019 | CNN | Steel | This method achieves significantly higher recognition accuracy for steel surface defects than state-of-the-art classifiers |
| [55] | 2019 | GAN | Steel | CAESGAN achieves the best classification rate compared to traditional methods, especially for hot rolled plates |
| [61] | 2019 | SDD and ResNet | Steel | Steel surface defect detection can be performed with high speed and accuracy |
| [62] | 2019 | Faster-RCNN | Steel | Achieved higher detection accuracy and more accurate location of defects, especially for tiny and slender defects |
| [63] | 2018 | CNN | DAGM dataset | Can achieve a 99.8% accuracy rate in detecting defects |
| [64] | 2016 | CNN | DAGM dataset | This method demonstrates a low false alarm rate and excellent defect detection results |

**Table 3.** *Cont.*

| Reference | Year | Method | Target | Performance |
|-----------|------|--------|--------|-------------|
| [65] | 2019 | FCN | DAGM dataset | A defect image (512 × 512) can be processed each second, with more than 99% of pixel accuracy |
| [66] | 2017 | 2-stage FCN framework | DAGM dataset | Able to achieve meaningful results in terms of performance and speed |
| [34] | 2016 | CNN | Texture | In comparison to traditional manual inspection systems, this method offers several advantages in time and cost savings |
| [67] | 2018 | AutoEncoder | Various materials | Compared to traditional hand-engineered feature extraction methods, this approach is more generic |
| [68] | 2020 | CNN | | On the datasets, it is possible to achieve 100% recall and high precision |
| [69] | 2021 | YOLOv5 | PCB | Can achieve a 0.7% mAP promotion on HRIPCB dataset |
| [70] | 2021 | YOLOv3 | PCB | The detection rate increases to 63 frames per second due to an increase in mAP of 92.13%. As a result, PCB surface defect detection has increased application prospects |
| [71] | 2021 | CNN | Flexible printed circuit boards (FPCBs) | Achieves 94.15% mean average precision (mAP) in comparison with existing surface defect detection networks |
| [72] | 2022 | CNN | Rails | Detected 98.2% of defects at the image level and 97.42% at the pixel level, respectively |
| [73] | 2021 | YOLOv3 | RailwayHub | High-speed rail wheels can be detected more accurately and many defects can be located with greater accuracy with this system |
| [74] | 2019 | Faster R-CNN | Railway insulator | Algorithms superior to others |
| [75] | 2017 | CNN and SVM | Metal surface | In classification, this method outperforms both state-of-the-art traditional handcrafted features and other deep ConvNet features extracted from a preselected best layer based on several anomaly and texture datasets |
| [76] | 2021 | CNN | Metal Workpiece | It has strong adaptability and is capable of automatically extracting and detecting defects |
| [77] | 2021 | YOLOv5 | Insulator | It reduces unsafe manual detection and increases detection efficiency by effectively identifying and locating insulator defects across transmission lines |
| [78] | 2021 | Mask R-CNN | Insulator | Detection accuracy: 87.5% |
| [79] | 2021 | SE-YOLOv5 | Fabric | As compared to the original YOLOv5, the improved SE-YOLOv5 has a higher accuracy, generalization ability, and robustness for detecting fabric defects |

**Table 3.** *Cont.*

| Reference | Year | Method | Target | Performance |
|---|---|---|---|---|
| [80] | 2021 | YOLOv4 | Fabric | Can quickly and accurately locate defects, and can also be used in other defect detection industries |
| [81] | 2022 | UNet | Fabric | Detection accuracy rate: 99% |
| [82] | 2022 | SVM | Non-woven fabric | It is highly accurate and performs well in real time |
| [83] | 2021 | Faster R-CNN | Aluminum | In comparison with the original algorithm, this algorithm achieved 78.8% mean average accuracy (mAP), which is 2.2% higher |
| [84] | 2018 | CNN | Copper clad lamination surface | Accuracy rate: 98.2% |
| [85] | 2019 | Faster-RCNN and feature fusion | (GAPR) texture defect dataset | Performs well under various conditions and has good adaptability |
| [86] | 2022 | Autoencoder and morphological operation | Textile | Superior performance to other prevailing models |
| [87] | 2019 | Faster R-CNN | Weel hub | It is simpler, faster, and more accurate than both R-CNN and YOLOv3 methods for wheel hub defects |
| [88] | 2022 | YOLOv3 | Polarizer | There is a slight increase in its mAP over YOLOv3, and it has a detect speed increase of 44% to 121 frames per second |
| [89] | 2021 | Faster R-CNN | Belt Layer of Radial Tire | False negatives and false positives decrease by 7.79%, 3.4%, and 5%, respectively, compared with the vanilla Faster R-CNN |
| [90] | 2017 | CNN | Pavement crack analysis | Accurately detects pavement cracks and evaluates their types |
| [91] | 2022 | YOLO v5 | Solar Cell | Solar cell EL images were used to train the model, which achieved 89.64% mAP |
| [92] | 2017 | CNN | Mangosteen | Recognition accuracy: 97% |
| [93] | 2017 | CrackNet | Crack detection on 3D asphalt surfaces | With 200 3D images, CrackNet achieved high precision |
| [94] | 2021 | R-CNN | Textile fabric | Defect detection accuracy improved by 4.09% to 95.43% |
| [95] | 2020 | CNN | AigleRN and DAGM2007 | Can achieve high detection accuracy and efficiency |
| [96] | 2019 | Faster R-CNN | Aluminum profile | With regard to the multiscale defect-detection network, it achieved a 75.8% mAP over Faster R-CNN |
| [97] | 2022 | MobileNetV3 | Sanitary ceramics | With the Faster R-CNN method, detection speed is improved by 22.9%, precision is improved by 35.0%, and memory consumption is reduced by 8.4% compared to the SSD, YOLO V3, and one-stage SSD methods |
| [98] | 2017 | CNN | Welding | Recognition accuracy rate: 95.83% |

**Table 3.** *Cont.*

| Reference | Year | Method | Target | Performance |
|---|---|---|---|---|
| [99] | 2017 | CNN | Concrete cracks | The CNN is trained on 40,000 images with a resolution of 256 × 256 pixels and achieves an accuracy rate of approximately 98% |
| [100] | 2022 | YOLOv5 | Plastic | Superior performance to other prevailing models |
| [101] | 2019 | SDD-CNN | Roller subtle | Accuracy rate: 99.56% |
| [102] | 2018 | GAN | MPCG (Mobile Phone Cover Glass) | MPCG defects can be detected with high accuracy of 98% |
| [103] | 2022 | YOLOv5 | Ceramic ring | Accuracy rate: 89.9% |
| [104] | 2018 | CNN | Solar cell | Recognition rate: 94.30% |
| [105] | 2022 | Wavelet Decomposition and CNN | Automobile Pipe Joints | Reduces the impact of uneven illumination, random noises, and texture processing on defect classification accuracy, and the SVM classification method demonstrates an accuracy of approximately 83% for identifying the presence of no defects, pits, and scratches in a given set of data |
| [106] | 2021 | Multi-Feature Fusion and PSO-SVM | Lithium Battery Pole Piece | Average recognition rate: 98.3% |
| [107] | 2018 | CNN | Shinny surfaces | Classification rate: around 89% |
| [108] | 2017 | DL-based ASI | NEU, Weld, and wood defect database | Can improve the accuracy by 0.66% to 25.50% for datasets |
| [109] | 2022 | SCED-Net | Steel Coil | As compared to recent networks used in steel coil end face detection and some classical object detection networks, this method offers better performance |
| [110] | 2021 | FFCNN consists of (feature extraction module, feature fusion module, and decision-making module) | Magnetic Tile | The performance of a combination of mean fusion and Resnet-50 with CBAM is 97.0%, while the combination of max fusion and Resnet-50 with CBAM has an accuracy rate of 95.0% |
| [111] | 2018 | AlexNet and SVM | Custom dataset | Detection Accuracy: 99.201% |
| [112] | 2021 | YOLOv3 | Chip | mAP REACHES 86.36% |
| [113] | 2017 | CNN and a voting mechanism | Metallic gasket, DAGM defects, and screw image | Performs well in arbitrary textured images as well as in images with special structures, proving that it is superior to traditional detection algorithms |
| [114] | 2022 | CNN | High Voltage Circuit Breaker | The network model has been shown to be able to accurately detect four different levels of rust through experimental results, with a success rate of 94.25% |

**Table 4.** Strengths and weaknesses of different techniques for detecting surface defects on industrial products using deep learning.

| Approach | Reference | Method | Strengths | Weaknesses |
|---|---|---|---|---|
| Supervised | [33] | Two-layer neural network | Able to detect cross-category defects without retraining; simplicity of the structure of the model allows for faster training and inference | Limited to only two layers; may not be able to extract complex features; the simplicity of the model may make it less robust to noise and other variations in the input data |
| | [34] | Composition of kernels | Efficient network architecture for detecting small defects and textures in surface images | Lack of emphasis on the number of layers may lead to suboptimal results |
| | [35,36] | ShuffleNet | Can only be trained with negative images | May not perform well on larger; more complex datasets |
| | [37] | Shallow CNN | Significantly improves detection speed and can be used for end-to-end learning | Limited to identifying anomalies and may not perform well on more complex defects |
| | [38] | Faster RCNN | Requires a separate region proposal network; significantly improves the speed of target detection; can detect objects of different scales. | Might not perform well on highly cluttered scenes with many overlapping objects. |
| | [39] | Cascaded RCNN | Can effectively solve the defect detection problem for specific applications such as power line insulators | May not perform well on defects with irregular or unpredictable shapes |
| | [9] | MobileNet-SSD | Highly efficient and capable of real-time object detection in limited hardware configurations | May not perform as well as other models on larger, more complex datasets |
| | [42] | FCN | Can achieve high accuracy and directly output label maps at the pixel-level | Can be computationally expensive, especially when used with large datasets |
| Unsupervised | [46] | PatchCore | Identifies and isolates abnormal data in scenarios where only normal examples are available | May not perform as well as other models on larger and more complex datasets |
| | [47] | DBN | Utilizes both training and reconstructed images as supervision data for fine-tuning; can learn useful features from the data without the need for manual feature extraction, which can save time and resources | May not have the capacity to identify more complex features in the images |
| | [48] | Multi-scale convolutional denoising autoencoder | High precision and robustness by combining results from multiple pyramid levels; can effectively remove noise from the input data, which can improve the performance of defect detection in noisy images | May not be able to generalize well to new unseen data, especially if the data is vastly different from the training data; computationally expensive to train, especially when the input data is high-dimensional, which can be a limitation in real-time applications |
| | [49] | SOM-based detection | Can effectively cluster and classify high-dimensional data, which can be useful for detecting defects in images and other types of data | Can be sensitive to the initial conditions of the map and the choice of parameters, which can make it challenging to obtain accurate and consistent results |
| | [50] | GANs | Two-stage process for detecting new areas and directly distinguishing defects and normal samples | GANs can be difficult to train and may require a large amount of data |

**Table 4.** *Cont.*

| Approach | Reference | Method | Strengths | Weaknesses |
|---|---|---|---|---|
| | [51] | Multiscale AE with fully convolutional neural networks | Obtains the original feature image and performs feature clustering through each FCAE sub-network; can effectively learn spatial relationships between pixels, which can be useful for detecting defects in images | May struggle with detecting small or subtle defects, which may not be easily distinguished from normal patterns in the input data |
| | [52] | GAN-based surface vision detection framework | Proven effective on datasets of wood cracks and road cracks; can be used to generate images that can be used to improve the interpretability of the model and help identify the specific features that are used to detect defects | May struggle to generate high-quality images if the training dataset is small or of poor quality; may face mode collapse problem, where the generator produces only a small subset of all possible outputs |
| | [53] | GAN-based method for detecting strip steel surface defects | Tailored for detecting strip steel surface defects, it could be more effective and accurate than general-purpose models | Performance may be limited to the specific application of detecting strip steel surface defects and may not generalize well to other types of defects or materials |
| Semi-Supervised | [54] | Active learning and self-training | Improves classification accuracy while reducing the need for manual annotations | Can be limited by the quality of the unlabeled data, which may contain a large number of examples that are not relevant to the task at hand |
| | [55] | Convolutional Autoencoder and Generative Adversarial Network | Allows the model to effectively extract high-level features from the input data, which can be useful for detecting defects | May struggle to generate high-quality images if the training dataset is small or of poor quality |
| | [56] | WSL framework | Combines localization networks and decision networks for effective detection of real industrial datasets | May not perform well on images with intricate backgrounds |
| | [58] | Semi-supervised learning system | Generates samples to detect surface defects with improved accuracy compared to supervised and transfer learning methods | May not perform well on images with intricate backgrounds |
| | [59] | Residual network structures | Shows good performance on DAGM, NEU steel, and copper clad plate datasets with a balanced network depth and width | May require more computational resources to train |

**Table 5.** A list of common surface defect datasets with classifications for industrial products.

| Name and Reference | Target | Link |
|---|---|---|
| MVTec AD [115] | Various materials | http://mvtec.com/company/research/datasets (accessed on 2 February 2023) |
| Steel Defect Detection | Steel | https://kaggle.com/c/severstal-steel-defect-detection/data (accessed on 2 February 2023) |
| GC10–Det [116] | Metal | https://kaggle.com/alex000kim/gc10det (accessed on 2 February 2023) |
| Industrial Metallic Surface Dataset | Metal | https://kaggle.com/datasets/ujik132016/industrial-metallic-surface-dataset (accessed on 2 February 2023) |
| Bridge Cracks [117] | Bridge | https://github.com/Iskysir/Bridge_Crack_Image_Data (accessed on 2 February 2023) |
| Fabric defect dataset | Fabric | https://kaggle.com/datasets/rmshashi/fabric-defect-dataset (accessed on 2 February 2023) |

**Table 5.** *Cont.*

| Name and Reference | Target | Link |
|---|---|---|
| DeepPCB dataset [118] | PCB | https://github.com/tangsanli5201/DeepPCB (accessed on 2 February 2023) |
| PCB Defects | PCB | https://kaggle.com/datasets/akhatova/pcb-defects (accessed on 2 February 2023) |
| PCB DSLR DATASET | PCB | https://zenodo.org/record/3886553#.Y1dNl3bMKUk (accessed on 2 February 2023) |
| Structural Defects Network (SDNET) 2018 [119] | Concrete | https://kaggle.com/datasets/aniruddhsharma/structural-defects-network-concrete-crack-images (accessed on 2 February 2023) |
| COncrete DEfect BRidge IMage Dataset | Concrete | https://zenodo.org/record/2620293#.Y1dPO3bMKUk (accessed on 2 February 2023) |
| Surface Crack Detection Dataset [120] | Concrete | https://kaggle.com/arunrk7/surface-crack-detection (accessed on 2 February 2023) |
| Pavement crack dataset | Pavement | https://github.com/fyangneil/pavement-crack-detection (accessed on 2 February 2023) |
| Cracks and Potholes in Road Images Dataset | Road | https://biankatpas.github.io/Cracks-and-Potholes-in-Road-Images-Dataset (accessed on 2 February 2023) |
| Crack Forest Datasets [121] | Road | https://github.com/cuilimeng/CrackForest-dataset (accessed on 2 February 2023) |
| Tianchi aluminum profile surface defect dataset | Aluminum | https://tianchi.aliyun.com/competition/entrance/231682/information (accessed on 2 February 2023) |
| Solar cell EL image defect detection | Solar panel | https://ieee-dataport.org/documents/photovoltaic-cell-anomaly-detection-dataset (accessed on 2 February 2023) |
| Elpv-dataset [122] | Solar panel | https://github.com/zae-bayern/elpv-dataset (accessed on 2 February 2023) |
| Magnetic tile surface defects [123] | Tile | https://github.com/abin24/Magnetic-tile-defect-datasets (accessed on 2 February 2023) |
| Dataset for Rail Surface Defects Detection | Rail | https://arxiv.org/abs/2106.14366 (accessed on 2 February 2023) |
| Railway Track Fault Detection | Rail | https://kaggle.com/datasets/salmaneunus/railway-track-fault-detection (accessed on 2 February 2023) |

## 4. Deep Learning Defect Detection Methods for X-ray Images for Industrial Products

Non-destructive testing (NDT) is a method that uses radiography or ultrasound technologies to discover faults without causing damage to the detected objects. It is widely used in engineering industries to detect and evaluate defects in materials of all types.

An important technique in non-destructive testing is radiographic testing, which uses X-rays to identify and evaluate flaws or defects, such as cracks or porosities. Defects can appear in X-ray images in many shapes and sizes, making detection difficult. The images are often low contrast and noisy, making identification of defects difficult.

The traditional approach for identifying defects in industrial products is for human operators or experts to visually inspect radiographs. However, this method can be subjective and prone to errors. Additionally, the process of examining a large number of images can be time-consuming and may lead to misinterpretations. However, there have been significant advancements in the field of defect detection in recent years, thanks to the emergence of deep learning techniques. As a result, a number of methods for detecting defects have been proposed, which are more efficient and reliable than the conventional approach. This section aims to provide a summary of current research on industrial product defect detection methods using X-ray images. Specifically, it covers the use of deep learning techniques such as convolutional neural networks and generative adversarial networks to analyze radiographic images and identify defects with a high degree of accuracy. These methods have the potential to reduce the subjectivity and human errors associated with the traditional approach, as well as the time required for inspection. Additionally, they can be trained to improve over time with more data, making them more robust and reliable.

A proposed system in literature [124] aimed to automate the process of inspecting and monitoring the condition of machines in the hard metal industry by analyzing defects in real production samples. Three models were created to analyze different types of data, a method called stacked generalization ensemble was applied and a random forest classifier was utilized to combine and analyze the results of the microprofilometer and ultrasound models. The fusion model was found to have improved performance and higher classification accuracy (88.24%) as compared to the individual models. Additionally, the shop floor model was able to effectively identify breakdowns during the manufacturing process and the ultrasound model was found to have better classification scores compared to the VGG-19 model. According to literature [125], a three-stage deep learning algorithm was proposed for detecting bubble patterns in engines. The algorithm consisted of training an autoencoder using normal images, fixing the coefficients of the encoder, and training a fully-connected network using both normal and defective images. To improve the performance of the network, the entire system was fine-tuned. According to [126], a CNN model was designed with ten layers that belong to six grades for detecting defects in X-ray welding images. It was possible to achieve 98.8% classification accuracy using CNN if the ReLU activation function was used for X-ray welding image recognition. A real-time X-ray image analysis method using Support Vector Machines (SVMs) was presented in [127]. Using a background subtraction algorithm, all potential defects were segmented, and three features were extracted, including the defect area, the grayscale average difference, and the grayscale standard deviation. In order to distinguish non-defects from defects, the extracted features were input into an SVM classifier. A real-time X-ray image defect detection method based on the proposed method reduced undetected defects and false alarms. Another SVM-based method for detecting weld defects was described in [128]. The training SVM is trained by extracting three feature vectors from potential weld defects using grey-level profile analysis. In the last step, the SVM is trained to differentiate between defects that are real and those that are potential. A high percentage of correct detections could be achieved using the proposed method. For detecting insert molding in automotive electronics, ref. [129] proposed a Yolov5-based DR image defect detection algorithm. Width and a window level are adjusted in the preprocessing stage of the acquired data, and fast guided filtering is used for edge retention. Using the overlap, tiny anomalies are detected, and a multi-task dataset is constructed. Using Ghost, which replaces the standard convolutional network with the backbone network with enhanced features, the number of parameters can be further reduced. Moreover, CSP-modules are embedded in the neck and backbone of the network to enhance feature extraction. As a result of adding the transformer attention module after spatial pyramid pooling, over-fitting can be avoided while computational effort can be reduced. DR data-based Yolo series target detection algorithms are used as a final step to conduct consistent experiments. For detecting bead toe errors, ref. [130] proposed a lightweight semantic segmentation network. An encoder extracts the texture features of different regions of the tire in the network first. Then, to fuse the encoder's output feature, a decoder is introduced. A reduction in the dimension of the feature maps has allowed the positions of the bead toe to be recorded in the X-ray image. An index of local mIoU (L-mIoU) is proposed to evaluate the final segmentation effect. YOLOv3_EfficientNet is used as the backbone of the methodology instead of YOLOv3_darknet53. It results in a substantial improvement in YOLOv3 mean average precision, as well as a substantial reduction in inference time and storage space. DR image features are then used to enhance the data, thereby increasing the diversity of the clarity and shape of defects. With depth separable convolution, models can be deployed on embedded devices with acceptable accuracy loss ranges. A method was presented in [131] that utilizes deep learning with X-ray images to detect defects in aluminum casting parts used in automobiles, with the goal of improving the accuracy of both the algorithm and data augmentation. The study found that using Feature Pyramid Networks (FPNs) resulted in a 40.9% increase in Mean of Average Precision (mAP) value, making it the most effective modification. Additionally, using RoIAlign instead of RoI pooling in Faster R-CNN improved the accuracy of bounding

box location. The study also proposed various data augmentation methods to compensate for the limited availability of X-ray image datasets for defect detection. The results showed that the mAP values for each data augmentation method reached an optimal value and did not continue to increase as the number of datasets increased. Overall, the proposed improvements to the Faster R-CNN algorithm resulted in better performance for X-ray image defect detection of automobile aluminum casting parts. Using the Faster R-CNN detection model with X-ray preprocessing was applied to the detection of tire defects in [132] to improve curve fitting performance. Faster R-CNN precision and recall of defects were improved by adjusting its feature extractor, proposal generator, and box classifier. According to literature [133], triplet deep neural networks can be used to detect weld defects. X-ray images are first preprocessed into relief images to make defects easier to identify. Following that, a deep network is constructed based on triplets, and a feature vector is obtained by mapping the triplets. The distance between similar defect feature vectors and the distance between different types of defect feature vectors must be closer. The SVM is also used for automatic detection and classification of weld defects. Based on the results of two experiments, the proposed method is capable of effectively detecting multiple defects. Tables 6 and 7 together provide a comprehensive overview of the current state of research and practices in the field of deep learning for defect detection in X-ray images. Table 6 summarizes recent research publications, and Table 7 compares the strengths and weaknesses of different techniques. This information can be valuable for anyone interested in the advancement of this field.

**Table 6.** Recent publications on deep learning defects detection in X-ray images.

| Reference | Method | Target | Performance |
|---|---|---|---|
| [125] | Three-Stage Deep Learning Algorithm | Engines | Accuracy rate: above 90% |
| [126] | Convolutional Neural Network (CNN) | Welding | Recognition accuracy can be more than 90% |
| [127] | Support Vector Machine (SVM) | Welding | Accuracy rate: 99.4% |
| [128] | Support Vector Machine (SVM) | Welding | Rate of detection is approximately 99.1% |
| [129] | Yolov5 | Insert Molding | Recognition accuracy: 93.6% |
| [130] | Lightweight semantic segmentation network | Tire | Achieved 97.1% mIoU and 92.4% L-mIoU for 512 × 512 input images |
| [131] | Faster R-CNN | Automobile casting aluminum parts | RoIAlign showed a significant improvement in the accuracy of bounding box location compared to RoI pooling, resulting in an increase of 23.6% accuracy under Faster R-CNN |
| [132] | Faster R-CNN | Tire | Compared with other methods, this method is capable of achieving a higher level of detection accuracy |
| [133] | Triplet Deep Neural Network | Welding | Can be more effective than traditional methods. |
| [134] | Deep Convolution Neutral Networks | Aluminum Conductor Composite Core (ACCC) | Can be effective in recognizing small and inconspicuous defects, with a 3.5% improvement in mean Average Precision compared to RetinaNet |
| [135] | Unsupervised Learning with Generative Adversarial Network | Tire | A tire X-ray dataset achieves 0.873 Area Under Curve (AUC) |

**Table 6.** *Cont.*

| Reference | Method | Target | Performance |
|---|---|---|---|
| [136] | R-CNN | Metal | Can eliminate time-consuming and inconsistent criteria while making judgments more efficient and accurate |
| [124] | Deep Neural Networks (DNNs) | Actual samples from the hard metal production industry | Indicates that the fusion model outperforms the separate models in terms of recall (100%), precision (60%), F-score (75%), and accuracy (88.24%) |

**Table 7.** Strengths and weaknesses of different deep-learning techniques for identifying defects in X-ray images.

| Reference | Method | Strengths | Weaknesses |
|---|---|---|---|
| [125] | Three-stage Deep Learning Algorithm | Ability to adapt to different types of patterns; the three-stage approach allows for more accurate and efficient detection of defects | The accuracy of the model can depend on the specific models used in each stage, if the models are not well-suited for the task, the performance may suffer |
| [126] | CNN model with 10 layers | Ability to achieve high classification accuracy | May not work well with other types of images |
| [127] | SVM-based method | Achieved real-time X-ray image analysis and reduced undetected defects and false alarms; can work well with small datasets | SVM's can be sensitive to the choice of kernel and parameters |
| [129] | Yolov5-based DR image defect detection algorithm | Ability to detect tiny anomalies and improve edge retention by using fast guided filtering | May not work well with other types of images or industries |
| [130] | Lightweight semantic segmentation network | The dimension reduction allows for accurate recording of bead toe positions in X-ray images; can be trained to work with different types of x-ray images, such as mammograms or chest x-rays | The model may not generalize well to different types of images |
| [131] | Deep learning with X-ray images and Feature Pyramid Networks (FPNs) | 40.9% increase in Mean of Average Precision (mAP) value, can effectively detect objects at different scales, which is important for defect detection in X-ray images as defects can be small and difficult to spot | May have a high false positive rate as X-ray images can have many benign structures that could be mislabeled as defects |
| [132] | Faster R-CNN detection model with X-ray preprocessing | Improved curve fitting performance; able to handle multiple defect classes; can handle images of different scales, which is important for defect detection in X-ray images, as defects can be small and difficult to spot | Limited to specific type of image and specific type of defect; may have a high false positive rate as X-ray images can have many benign structures that could be mislabeled as defects |
| [133] | Triplet deep neural network | Effective at detecting multiple defects, it works well with X-ray images, by preprocessing them into relief images to make defects easier to identify | It may not generalize well to different types of images |
| [124] | Stacked Generalization Ensemble | Improved performance and higher classification accuracy compared to individual models; ability to effectively identify breakdowns during manufacturing process; the ensemble approach can improve the robustness of the model by combining the strengths of multiple models | May not work well with other industries or types of defects |

## 5. Problems and Solutions

### 5.1. Unbalanced Sample Identification Problem

In industrial products, surface defects can also be detected with deep learning using unbalanced sample sets [137,138]. To train the deep learning model, it is usually necessary to have a balanced sample set of samples of different categories. This ideal situation, however, almost never occurs in the real world. More often than not, the majority of data in the dataset comes from "normal" samples, while "defective" or "abnormal" samples only make up a small portion. Supervised learning is one of the main tasks that suffers from unbalanced sample identification. The algorithm will therefore pay more attention to categories with larger data volumes and underestimate categories with smaller data volumes, affecting the model's generalization and prediction abilities. The data-level process methods aim to maintain a consistent number of samples for all types within the training set. Resolving the unbalanced sample identification issue at the data level can be broken down into five categories: data resampling, data augmentation, class equalization sampling, data source, and synthetic sampling. It is necessary to collect more samples in fewer categories from the data source. By horizontally or vertically flipping, rotating, zooming, cropping, and other operations, we can purposefully increase the number of sample data in each category.

Regarding data resampling [139,140], it is good to resample a sample set to change the proportion of samples in each category, including oversampling and undersampling. Class equalization sampling groups samples by categories and generates sample lists for each category. To ensure that each category has an equal chance of participating in training, a random category is selected during training, and samples are randomly selected from the corresponding sample list. Synthetic samples [141] are generated by combining various characteristics of an existing sample to create a new sample. Using this method, you can create a new sample by randomly selecting a value from the feature.

### 5.2. Small Sample Problem

As a result of continuous optimization of industrial processes, the number of defective samples has decreased. This makes it difficult to use deep learning methods to detect surface defects in industrial products, since there are fewer and fewer defect images available for deep learning. Overfitting problems in training can easily occur with small samples [142]. Transfer learning applies knowledge gained from one task to a different but related task when there is insufficient data to complete the target task. Consequently, transfer learning is also a critical method for solving the small sample problem. For surface defect detection, literature [143,144] used VGG networks and transfer learning to detect emulsion pump bodies, printed circuit boards, transmission line components, steel plates, and wood surfaces. Fabric surface defect detection using DenseNet and transfer learning was described in [145]. The combination of transfer learning and AlexNet was used to detect surface defects on solar panels and fabrics in [146,147]. Solving the small sample problem can also be achieved by optimizing the network structure. For the first time, GAN was used for image anomaly detection with the AnoGAN model [148] in 2017. A continuous iterative optimization process is used to find an image that matches the test image closest in the latent space, and then DCGAN is used to detect anomalies in that image. The f-AnoGAN model was introduced in [149]. This model proposes a method of encoding an image so that latent points can be quickly mapped, and then using WGAN to detect anomalies. As a result of the introduction of an encoder, the AnoGAN's iterative optimization process is much faster and less time-consuming. Additionally, the GANomaly model was proposed in [150] in 2018. It detects abnormal samples by comparing latent variables obtained by coding with latent variables obtained by reconstructing. There is no requirement for training with negative samples in any of the above models. It is also possible to obtain many sample images by enlarging the data. Using synthetic defects [151], the decorated plastic part dataset is expanded by adding synthetic defects to the defect-free

image. Literature [152] described a technique for generating defect representations that combine hand-made and unsupervised learning features.

### 5.3. Real-Time Problem

It is essential to consider real-time problems when performing surface defect detection in real industrial environments. Real-time detection problems involve reducing detection time and improving detection efficiency to maintain roughly the same accuracy. Research has been conducted on real-time problems by some scholars. To detect surface defects on printed circuit boards, literature [153] suggested combining SSIM and MobileNet. Comparing the proposed algorithm with Faster R-CNN, it maintained high accuracy while being at least 12 times quicker than the existing algorithm. Literature [154] developed a novel 11-layer CNN model for detecting welding defects in robotic welding manufacturing. The proposed method was capable of detecting metal additive manufacturing in real time, which meets specific requirements for online detection.

## 6. Discussion

Deep learning technology has revolutionized the field of defect detection in industrial products. However, finding a suitable deep learning model for solving the defect detection problem is very difficult due to the particularities of industrial scenarios. In the coming years, deep learning will encounter challenges and trends as it becomes more widely used in industrial fields. A brief description of recent trends and future research directions is provided in this section.

- Integrating deep learning with other methods:

  By incorporating other techniques such as traditional image processing, the robustness and performance of the defect detection system in challenging conditions can be enhanced. For instance, using traditional image processing techniques to preprocess the images before inputting them into a deep learning model can improve the quality of the data and make it easier for the model to effectively detect defects. Additionally, integrating deep learning with other techniques, such as physics-based simulations, can provide better understanding of the underlying physical causes of defects and lead to the development of more efficient and effective defect detection methods.

- Adjustment to various lighting scenarios:

  Examining industrial products frequently occurs under diverse lighting conditions, which can make it hard to identify defects. Research in this field could concentrate on developing techniques for adapting to various lighting conditions and using them to enhance the precision of defect detection. This could include methods such as image enhancement techniques, color constancy techniques, and multiple exposure fusion techniques, to improve the visibility of defects in different lighting conditions. Additionally, research could also focus on developing deep learning models that are robust to changes in lighting conditions, such as using adversarial training methodologies, to improve the robustness of the model. This may lead to a more accurate and reliable defect detection system that can function in a wide range of lighting scenarios.

- Transparent AI:

  To be implemented in industrial environments, defect detection systems need to be transparent and explainable. Research in this field could focus on developing techniques to make deep-learning-based defect detection systems more understandable, so that users can comprehend why a defect was missed or incorrectly identified.

- All aspects need to be taken into account:

  In order for a defect detection system to perform well, it must take into account various factors. There are many factors that can influence the accurate detection of defects, such as defect size, shape, the technique for image acquisition, alignment and distortion of images,

resolution of images, and algorithmic speed, among others. It is important to consider all of these factors when creating a mature and successful method.

- Limited number of defect samples:

    In many industrial applications, deep learning methods require a large training dataset and have high computational costs, and the number of defect samples is often insufficient to detect defects. Additionally, as the product line is frequently updated, new defect types are introduced and the detection process becomes more challenging. When training on normal samples, a simple defect detection method does not have any issues dealing with a small defect dataset, but, when it comes to defect localization and classification, the size of the dataset containing defects can be a challenge.

- Utilizing transfer learning:

    Defect patterns may be shared between two different application domains. There may be similarities in the morphology of cracks in two different materials, but they may be different in their sizes and colors. It is currently necessary to train two different networks in order to use current approaches. A well-trained, tested network can transfer its knowledge to a new network to speed up the training process. Currently, transfer learning is not effectively utilized in most approaches.

- Multi-modal sensor integration:

    Defect detection in industrial products often relies on visual inspections using cameras or other imaging devices. However, incorporating other types of sensors, such as thermal, acoustic, or vibration sensors, can provide additional information that can aid in the detection of defects. Research in this area could focus on developing methods for integrating data from multiple sensors and using it to improve the accuracy of defect detection. This could include techniques such as sensor fusion, where data from multiple sensors is combined to provide a more comprehensive view of the product, or methods for combining deep learning with other types of sensor data, such as sensor data from IoT devices.

- Continuous learning:

    In industrial environments, the product line is frequently updated, and new defect types are introduced. Research in this area could focus on developing methods for continuous online learning, which can be used to adapt the defect detection system as new data is acquired and new defects are introduced. This could include online learning techniques, where the system can continuously update its knowledge as new data is acquired, or active learning methods, where the system can actively select the most informative images for annotation. This would allow the system to adapt to changes in the product line and improve its performance over time.

- Real-time detection:

    There are only a few existing defect detection methods that are implemented in real time. In order to apply these methods to real-time inspection scenarios in the future, computationally efficient methods must be developed among these methods in order to achieve detection success rates in real time.

- Reducing the complexity:

    Users of defect detection methods are interested in understanding why a defect has been missed or incorrectly identified in an acceptable part when such a method fails to find the defect. The majority of deep learning methods follow a complex architecture, so humans have difficulty understanding the decision-making process and providing a rationale for failure. When it comes to deploying and improving the performance of a system, this can be a challenge. Moreover, in industrial applications, lightweight deep learning networks will be easier to deploy. Often, the processing resources used to support artificial intelligence computations are valuable in quality inspections on production lines and industrial maintenance monitoring. By using lightweight networks, the prediction

system's workload can be effectively reduced, which is extremely beneficial for simple terminal deployments and can also reduce costs and performance.

- A common reference database:

Testing can be conducted on different databases, though several studies have failed to provide satisfactory results due to inconsistency in such databases and a lack of testing samples. Additionally, most of the studies presented in this review have their own databases with varying sizes and quality. To evaluate and compare performance in the future, a common reference database would be helpful.

## 7. Conclusions

Deep learning is rapidly gaining momentum as a powerful tool in the field of defect detection on industrial products. In this paper, we conducted a comprehensive review of the current state-of-the-art in the use of machine learning methods for detecting defects in industrial products. We specifically focused on deep learning methods for detecting surface defects and defects from X-ray images, and provided a detailed overview of the different techniques and algorithms that have been proposed in these areas. We also discussed some of the key challenges and limitations of these methods, and highlighted potential solutions to these problems. The goal of this review was to provide researchers with a clear understanding of the current state-of-the-art in the field of surface defect detection for industrial products, and to serve as a reference for future research in this area.

**Author Contributions:** Authors contributed as follows: Conceptualization, A.S. and J.R.; methodology, J.R. and M.E.-G.; funding acquisition, J.R. and M.E.-G.; investigation, A.S., J.R. and M.E.-G.; writing original draft preparation, A.S. and J.R.; writing—review and editing, A.S., J.R. and M.E.-G.; supervision, J.R. and M.E.-G. All authors have read and agreed to the published version of the manuscript.

**Funding:** This research was funded by Natural Sciences and Engineering Research Council of Canada (NSERC), grant number 210471.

**Data Availability Statement:** In the manuscript, you will find a list of the corresponding websites.

**Conflicts of Interest:** There are no conflicts of interest between the authors.

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
