# Peer review of "Defect Detection Methods for Industrial Products Using Deep Learning Techniques: A Review"

_algorithms, doi:10.3390/a16020095_

Round 1

Reviewer 1 Report

This is an interesting review paper covering defect detection for industry applications. Overall the paper is comprehensive, in my opinion though the following points should be addressed before publication.
1. The first sentence of paragraph 2 in the second page should be revised (lines 56 - 59), it is quite difficult to follow.
2. In lines 129 to 146, where other review papers are included, I believe the work of Papageorgiou et al. "Short Survey of Artificial Intelligent Technologies for Defect Detection in Manufacturing" should be included as it is highly relevant.
3. In Section 2, Table 2, and in Section 3, Table 4, please include the accuracy metric for all methods so that the reader has a clear picture on the performance of each method and the saturation of the used datasets.
4. In Section 2, the work of Roth et al. "Towards Total Recall in Industrial Anomaly Detection" should be included.
5. In page 9, line 273, the "literature [ICCS]" part should be revised.
6. In Section 3, the work of Kotsiopoulos et al. "Deep multi-sensorial data analysis for production monitoring in hard metal industry",  should be cited in the beginning where ultra-sound is mentioned (line 294) as it combines ultra sound and laser-based surface defect detection using DL.
7. A careful proofread is recommended, for instance "and In order", "addition, A wide range", "algorithm. width" .....

Author Response

The response to the reviewers is attached.

Reviewer 2 Report

This is a review article analyzes the current state of research to detect defects 16 using machine learning methods. The methodology is clear, the analysis is thorough with a great number of surveyed articles in international literature (166). The article is well written with a very good use of language. Categorization of relevant literature is sound and the article covers a significant number of different manufacturing paradigms, as well as different methods for obtaining defect data. Also, in discussion, the article presents ways found in literature that improve recognition of these defects.

A table of Abbreviations should be included. Check the Funding section, because it is incomplete.

Figure 2 legend should be revised, (a) and (b) are mixed up.

Author Response

File is attached.

Reviewer 3 Report

The paper, "Defect Detection Methods for Industrial Products Using Deep Learning Techniques: A Review," is a simple list of names of different neural models; it cites technical literature and provides examples of their application to nondestructive diagnostics in an extremely concise manner. It does not give any indication of the strengths or limitations of the different models listed, which would, on the contrary, be extremely useful in helping the reader in the possible choice among the different paradigms where he has to use them for applications of his interest. The indications proposed in the "discussion" are absolutely generic and do not alter the judgment that the work is of little use.

I consider this work of very little relevance and do not think it deserves to be published.

Typographical typographical errors, such as those already present in the last lines of the abstract, or at line 147, or 174 just to enumerate the first ones, turn out to be of relative importance in this framework. Although this aspect is, per se, of small importance, it is significant of the care taken in the editing of the manuscript.

Author Response

File is attached

Round 2

Reviewer 1 Report

The authors have covered all my comments. Nothing more to add.

Reviewer 3 Report

The authors responded adequately enough to this reviewer's solicitations, introducing some comments on the limitations and potential of the different methods, thus allowing them to be compared.

The answers given to the other reviewers also contribute in the same direction to an improvement of the work, which is in this form useful and therefore worthy of publication.